# Fabrication and Characterization of Three-Dimensional Microelectromechanical System Coaxial Socket Device for Semiconductor Package Testing

**DOI:** 10.3390/s23146350

**Published:** 2023-07-12

**Authors:** Tae-Kyun Kim, Jong-Gwan Yook, Joo-Yong Kim, Yong-Ho Cho, Uh-Hyeon Lee

**Affiliations:** 1Department of Electrical and Electronic Engineering, Yonsei University, Seoul 03722, Republic of Korea; 2PMT (Protec MEMS Technology) Inc., Asan-si 31413, Republic of Korea

**Keywords:** coaxial socket, 3D MEMS structure, high-speed testing, chip package test, impedance matching

## Abstract

With the continuous reduction in size and increase in density of semiconductor devices, there is a growing demand for contact solutions that enable high-speed testing in automotive, 5G, and artificial intelligence-based devices. Although existing solutions, such as spring pins and rubber sockets, have been effective in various applications, there is still a need for new solutions that accommodate fine-pitch, high-speed, and high-density requirements. This study proposes a novel three-dimensional microelectromechanical system spring structure coaxial socket for semiconductor chip package testing. The socket design incorporates impedance matching for high-speed testing and addresses the challenges of fine-pitch and high-density applications. Mechanical tests are conducted to evaluate the durability of the structure and electrical tests are performed to verify electrical characteristics by utilizing a vector network analyzer up to 60 GHz. Our results have revealed promising performance and will help in further optimizing the design for potential production in the field and industry.

## 1. Introduction

The increasing demand for multifunctional and high-performance electronic products, including those used in the internet of things, artificial intelligence (AI), and 5G applications, has driven the development of high-density packaging technology to meet industry requirements. Consequently, the specifications for high-performance testing technologies have reached unprecedented levels. Moreover, existing fine-pitch contact solutions suffer from limitations in test bandwidth due to various factors, such as parasitic components, self-inductance, and interference [1,2].

A package device undergoes a mass production test process involving the utilization of a test socket and test printed circuit board (PCB). The test PCB plays a crucial role in supplying power and signals to the package under test (PUT) through the test socket. Initially, the primary objective of the mass production test is to validate the establishment of physical interconnections. However, as the data rate increases, the presence of parasitic elements in the test socket and PCB begins to impact the test reliability [3,4].

Currently, two types of test sockets are commonly used in the industry: a spring-type socket (known as pogo) and rubber socket. The pogo socket utilizes metal springs and contactors to establish electrical connections, while the rubber socket employs elastic polymers and conductive powders for conductivity [5,6,7,8,9,10,11]. However, both sockets face challenges in meeting high-density and fine-pitch requirements for high-bandwidth package testing. Furthermore, rubber sockets also suffer from durability issues due to prolonged exposure to high temperatures. To address these limitations, this paper proposes a novel solution: a three-dimensional (3D) microelectromechanical system (MEMS) coaxial socket. This socket, which is fabricated using the 3D MEMS process, features a coaxial structure with a core signal and outer ground shield, effectively mitigating electric field interference. Precise impedance and high-frequency signal quality are achieved by strategically positioning the core signal at a specific distance from the ground shield, leveraging dielectric properties. The main objective is to introduce the 3D MEMS coaxial socket as a high-performance solution for package testing by surpassing the limitations of existing socket technologies. The implementation and advantages of this innovative design are depicted in Figure 1.

The test socket is a specialized device designed to securely hold and connect a packaged integrated circuit (IC) during testing. It comprises a structure with signal paths that align with the electrical contacts of the IC. The socket may have a compression mechanism or other means to ensure proper contact between the IC and the signal paths in the socket. The packaged IC is carefully positioned within the socket, ensuring alignment of the electrical contacts with the corresponding signal paths in the socket structure. The compression mechanism is used to establish reliable electrical contact between the IC and the signal paths in the socket. This mechanism involves applying controlled pressure or force to the IC, pressing it against the signal paths, which results in contact with the corresponding signal paths in the socket. This establishes an electrical connection between the IC and the testing equipment.

Once the electrical connection is established, the test equipment can send electrical signals to the IC through the signal paths in the socket. Various electrical tests are performed, such as measuring voltage, current, and other device characteristics. During a test, the socket ensures that the electrical signals are transmitted accurately between the IC and the test equipment. This minimizes signal loss and interference, enabling a reliable evaluation of the IC performance and characteristics. The test socket was designed to provide repeatable and consistent electrical connections, allowing multiple tests of ICs without compromising accuracy or reliability.

Figure 2 provides a visual representation of the contact process. Initially, the 3D MEMS structure is in a normal state, suspended vertically with the assistance of polydimethylsiloxane (PDMS) as the dielectric material, ensuring its stability. When the package applies pressure on the socket, the structure undergoes compression, resulting in the shortening of signal-transmission paths. This unique property of the 3D MEMS structure allows efficient signal transfer. Moreover, the 3D MEMS structure exhibits excellent contact resistance, which is attributed to its specific configuration and the use of an electroplated metal alloy.

The socket comprises a 3D MEMS structure and a ground metal conductor, both enclosed within PDMS (a dielectric material with a relative permittivity of 2.63), forming the socket structure. Figure 3 presents a comparison of the structures of three different types of sockets. As illustrated in Figure 3a, the 3D MEMS socket represents an innovative approach that integrates MEMS technology with a coaxial structure for package testing. This advanced socket solution provides distinct advantages over conventional sockets, including precise impedance control, high-frequency signal quality, shortened signal-transmission paths, customizable design capabilities for high-density and fine-pitch applications, enhanced durability and longevity, and compatibility with a wide range of packaged ICs.

In Figure 3b, the rubber socket is characterized by its flexibility and ability to conform to various shapes and sizes of ICs. It offers a lower contact force compared to pogo sockets, which reduces the risk of damage to delicate ICs. However, its durability and lifespan are limited owing to potential degradation over time, particularly under high temperatures. Thus, achieving precise impedance control and high durability with rubber sockets remains a challenge [9]. Meanwhile, as shown in Figure 3c, pogo sockets utilize metal springs and contactors to establish electrical connections with the IC. The metal springs generate a force to ensure proper contact when a tip makes contact with the solder ball bump. Pogo sockets have a simple and reliable construction and are also suitable for applications requiring moderate contact force. They can accommodate a wide range of IC sizes and pitches [5]. However, they still have limitations in achieving precise impedance control because of the physical properties of the metal springs. Table 1 provides an overview and summary of three types of coaxial sockets, presenting a comparison of their advantages and disadvantages. The table also includes references for further information on each type of socket.

## 2. Proposed 3D MEMS Coaxial Socket

### 2.1. Design of 3D Coaxial Socket

A 3D MEMS coaxial socket was designed by utilizing computer-aided design (CAD) and SolidWorks. Prior to fabrication, we thoroughly evaluated the coaxial structure using the 3D electromagnetic (EM) simulator (ANSYS HFSS), carefully considering the feasibility of MEMS process parameters, such as structure line and space resolution. In addition, achieving a characteristic impedance of 50 Ω is crucial for minimizing signal reflection and degradation during signal transmission between the tested package and the test PCB. The optimal design that fulfilled all these conditions is shown in Figure 4.

The design of the 3D MEMS coaxial socket includes important structural parameters, such as head height, coaxial outer conductor shield, total height, and pad pitch as depicted in Figure 5. The signal structure is supported by PDMS, ensuring proper isolation without short circuits, while the ground structures are connected to the outer shield conductor, forming a bridge. The dimensions of the 3D MEMS coaxial socket are given in Table 2. To minimize crosstalk and maintain signal integrity, adjacent single-ended channels are shielded by a ground shield. The proposed 3D MEMS structure exhibited a pitch of 0.60 mm, a head height of 0.20 mm, an inner diameter of the outer conductor = 0.35 mm, an outer diameter of the inner conductor = 0.28 mm, and a total height of 0.70 mm.

### 2.2. Impedance Matching of the 3D MEMS Coaxial Socket

Conventional coaxial theory refers to the principles and concepts related to the design and operation of coaxial cables and connectors. A coaxial cable comprises two concentric conductors separated by an insulating material and enclosed within an outer conductor. The inner conductor carries the signal, while the outer conductor serves as the ground or return path. The key concept in coaxial theory is transmission line theory, which describes the propagation of electrical signals along the length of the coaxial cable. According to this theory, the coaxial cable can support the transmission of high-frequency signals with minimal loss and interference. The behavior of a coaxial cable is governed by various parameters. One important factor is the characteristic impedance ($Z_{0}$) of the cable, which represents the impedance presented to the signal and determines the efficiency of signal transmission. The characteristic impedance is determined by the geometrical dimensions of the inner and outer conductors as well as the dielectric material between them [12,13,14].

The theory of the 3D MEMS coaxial socket revolves around its design and functionality. The structure is designed to provide precise impedance control and high-frequency signal quality for package testing applications. It incorporates a coaxial structure consisting of a core signal and an outer ground shield. The core signal is positioned at a certain distance from the ground shield to achieve the desired impedance and minimize signal degradation. The 3D MEMS coaxial socket operates according to the principles of coaxial transmission lines. In a coaxial structure, the inner conductor carries the signal, while the outer conductor serves as the ground reference. This configuration aids to minimize EM interference and signal loss. By maintaining a specific dielectric constant and dimensions, the socket ensures proper signal transmission and impedance matching.

The characteristic impedance ($Z_{0}$) of a coaxial cable can be calculated using the following equation:(1)$$Z_{0}=\frac{138}{\sqrt{\varepsilon_{r}}}log_{10}\frac{D}{d}$$
(2)$$C=\frac{7.354\varepsilon_{r}}{log_{10}\left(\frac{D}{d}\right)}$$
(3)$$L=140log_{10}\left(\frac{D}{d}\right)$$
where $Z_{0}$ and $\varepsilon_{r}$ represent the characteristic impedance and relative permittivity of the dielectric, respectively, and *D* and *d* correspond to the inner diameter of the outer conductor and the outer diameter of the inner conductor, respectively. When the relative permittivity ($\varepsilon_{r}$) is held constant, the impedance of the system is primarily determined by the geometric parameters *D* and *d*. By utilizing MEMS processes, *D* and *d* can be precisely controlled and manipulated, thereby enabling effective impedance control within the system.

Figure 6b illustrates the variation in the outer shield and core structure dimensions; specifically, the ratios of the inner diameter of the outer conductor over the outer diameter of the inner conductor are 3.25, 2.75, 2.25, and 1.75, respectively. Furthermore, regarding the dimensions of the packaged device, the structure of the socket can be modified to accommodate the inner core size while considering the mechanical characteristics of the structure. This allows customization and optimization of the socket design based on the specific requirements of the package device. A simulated study was conducted to investigate the variation in the *D/d* ratio, which represents the inner diameter of the outer conductor. The analysis began with time-domain reflectometry (TDR) to measure the reflections resulting from signal propagation through 3D MEMS structure transmission. TDR sends a pulse through the medium and compares the reflections from the unknown transmission environment with those produced by a standard impedance. Figure 7a presents a comparison of the impedance for different *D/d* ratios in the 3D MEMS structure. It was observed that the impedance can be controlled by adjusting the *D/d* ratio, which allows predictable electrical performance. Furthermore, this study analyzed the insertion loss and return loss of the 3D MEMS structure. The distance between the signal core and the outer conductor GND introduces impedance mismatch, leading to degradation in the insertion loss and return loss, as depicted in Figure 7b,c, respectively.

### 2.3. Fabrication of the 3D MEMS Coaxial Socket

In this section, the fabrication process to verify the 3D MEMS coaxial socket is described. Research on microsystem technology has emerged significantly, offering a multitude of technological solutions. Notably, advancements in MEMS have been achieved by leveraging micromachining techniques and tools traditionally employed in the silicon (IC) industry. Ongoing MEMS research is focusing on integrating innovative materials and fabrication techniques with conventional semiconductor processes to develop a wide array of microfunctional devices capable of sensing, actuating, computing, and more. Promising applications for MEMS are anticipated in domains such as telecommunications, chemical analysis, and biomedical instrumentation. The realm of the microworld presents exciting challenges and opportunities for this new century [15].

In the fabrication of the 3D MEMS coaxial socket, the base metal structure comprised a Ni–Co alloy [14], which was electroplated using MEMS techniques. As shown in Figure 8, a substrate was utilized as the foundation. The process began by creating the base layer and performing lithography to define the desired pattern. Subsequently, an etching process was employed to clean the surface, followed by electroplating. This sequence was repeated for the first and second middle bumps. Electrodeposition offers the advantage of producing deposits with customized structures. This method is characterized by its affordability, convenience, ability to work under near-ambient conditions, and high reproducibility. Moreover, electrodeposition enables the plating of a diverse range of materials, including pure metals, alloys, and composites, onto various substrates [16]. Following the completion of the entire structure, a strip process was conducted to remove the sacrificial layer. Gold (Au) coating was applied to cover the entire surface, thereby enhancing the contact resistance. Subsequently, PDMS was poured into the socket to maintain and accurately position the signal structure, effectively preventing any potential shorting to ground. The incorporation of PDMS plays a vital role in the overall functionality and performance of the 3D MEMS socket.

The material properties of PDMS are summarized in Table 3 [17]. PDMS-based elastomers are extensively used for soft lithographic replication of microstructures in microfluidic and microengineering applications. Elastomeric microstructures are commonly required to fulfill explicit mechanical roles, and accordingly, their mechanical properties can critically affect device performance. The advantageous physical and chemical properties of PDMS have made it a popular choice in various biomedical applications, including contact lenses, catheters, prostheses, implant coatings, and base materials for microelectrodes. PDMS has also been used as a carrier for cells, such as in scaffold materials for bio-artificial livers and neuronal tissue reconstruction. Additionally, PDMS has found utility in micropumps and deformable biomembranes that mimic the motion of jellyfish, both activated by cardiomyocytes. PDMS is subjected to a range of mechanical loading conditions, including large deformations, monotonic and cyclic loading, and uniaxial and multiaxial stress states, in many of these applications [18,19]. In this study, the specific type of PDMS adjusted at the 3D MEMS coaxial socket was Sylgard 184, which comprised a base polymer and a crosslinking agent mixed in ratios prescribed by the manufacturer. The mixing ratio of 10:1 (base polymer to crosslinker) has been extensively used in the field. By employing this fabrication approach, the proposed MEMS process successfully achieved the desired 3D spring structure for the coaxial socket device.

## 3. Mechanical Charactersitics

### 3.1. Behaviour of 3D MEMS Structure

After fabrication of the 3D MEMS coaxial socket, a test was performed to evaluate the behaviour of the 3D MEMS structure upon pressure. Throughout the compression process, the resulting shape matched expectations precisely. As an innovative aspect of the design, a stroke of 0.2 mm was achieved, providing the desired range of motion for the structure. As shown in Figure 9, the structure undergoes compression of 0.1, 0.15, and 0.2 mm. The structure restored its original form after each overdrive.

### 3.2. Three-Dimensional MEMS Structure Force and Contact Resistance

This section presents a comprehensive analysis of the force and contact resistance testing of the 3D MEMS structure. To evaluate the contact force, a specialized test was conducted using the MEMS structure with an incremental overdrive of 10 µm up to a maximum of 200 µm. To measure the structure force accurately, the measurement system incorporated a load cell and contact tip. The load cell serves as a force transducer, converting various types of forces, such as tension, compression, pressure, or torque, into an electrical signal that can be measured and standardized. As the tip pushes against the MEMS structure, the load cell detects and translates the applied force into a proportional electrical signal. This measurement system enables precise analysis of the structure force, providing valuable insights into the mechanical behavior of the 3D MEMS socket. Simultaneously, the contact resistance during each increment was measured.

The relationship between contact force and contact resistance is depicted in Figure 10. As the applied overdrive was increased up to 200 µm, the contact force exhibited a corresponding rise, while the contact resistance showed stability and a gradual decrease. Notably, the contact resistance reached saturation at approximately 120 µm, with a recorded value of approximately 20 mΩ. This analysis provides valuable insights into the structure force and contact resistance characteristics of the 3D MEMS socket.

## 4. Electrical Characteristics

### Thermal Shock Test

A thermal shock test was conducted to evaluate the durability and productivity of the proposed 3D MEMS coaxial socket. The objective of the test was to evaluate the performance of the MEMS structure and the properties of the PDMS material under extreme temperature conditions. By exposing the socket to rapid temperature changes, its capability to endure thermal stress while preserving functionality was investigated. Thermal shock resistance refers to the ability of a solid material to withstand sudden and drastic temperature changes during heating or cooling processes. The evaluation of thermal shock resistance involves subjecting the material to cycles of specific temperatures within its working range.

In order to conduct the measurement, a carefully arranged test sample was prepared. The 3D MEMS structure coaxial socket sample was positioned onto a dummy PCB that featured microstrip lines, enabling accurate measurement of the resistance and ensuring proper contact between components. A pair of PCBs were specifically designed and fabricated for this purpose. The 3D MEMS structure was then compressed between these two pairs of PCBs, applying a controlled force of up to 180 µm. This compression was achieved using an inside seam ring, which was chosen for its compliance with the maximum stroke requirement, ensuring robustness in the experimental setup. The assembled configuration of the PCBs and the 3D MEMS socket structure provided stability and sustained resistance, ensuring reliable measurements. The setup was designed to withstand external disturbances and maintained its stability throughout the measurement process as shown in Figure 11.

To achieve rapid temperature changes, the device under testing (DUT) was placed inside a basket that could swiftly move between hot and cold zones in a matter of seconds. The temperature of these zones could be controlled using either an air-to-air or a liquid-to-liquid system. The resistance of a metal sample with known length and area could be measured using appropriate techniques. A basic ohmmeter utilizes two contacts, one at each end of the sample, to determine resistance; however, more precise measurements can be obtained by employing a four-contact device [20,21].

In this study, the thermal shock resistance of the 3D MEMS coaxial socket was assessed through the thermal shock cycling test to evaluate its resistance to rapid and extreme temperature changes. The test involved subjecting the socket to a temperature range of −45 °C to 85 °C for 30 min, completing one cycle in 1 h. A total of 500 cycles, equivalent to approximately 21 days, were performed using the ESPEC TSD-101 thermal shock chamber [22].

Real-time resistance measurements were performed using the Keithley 2602B source meter and HIOKI RM3545 resistance meter to determine the contact resistance of the 3D MEMS coaxial socket. The repeatability and reproducibility of the MEMS socket were assessed through thermal shock cycling tests. At a hot temperature of 85 °C, the maximum contact resistance was measured to be 0.1429 Ω, while at a cold temperature of −45 °C, the minimum contact resistance was observed to be 0.0906 Ω as illustrated in Figure 12. Prior to subjecting the socket to thermal cycling, the self-resistance of the dummy test PCB, which featured microstrip transmission lines made of FR4 material, was measured. The measurements revealed a self-resistance of 0.0611 Ω at 85 °C and 0.0395 Ω at −45 °C. Subtracting the PCB self-resistance temperature from the total resistance temperature, the maximum contact resistance of the 3D MEMS coaxial socket at a high temperature was 0.0287 Ω, while the minimum contact resistance at a low temperature was 0.0166 Ω.

Observations indicated that the contact resistance decreases at colder temperatures and increases at higher temperatures, aligning with the expected behavior. These results provide strong support for the feasibility and suitability of the proposed MEMS socket. The low-level contact resistance test conducted on the 3D MEMS coaxial socket played a crucial role in contributing to the overall findings and validation of the socket for its intended application. The results of the thermal shock test provided valuable insights into the durability and productivity of the MEMS structure and the behavior of the PDMS material across a wide temperature range.

Overall, the thermal cycle test was conducted to assess the durability and repeatability of the 3D MEMS structure, which is a crucial aspect for its application in industry standards. The results of the test demonstrated successful performance, with the contact resistance exhibiting excellent stability and compatibility over a period of 21 days. This evaluation provides valuable insights into the reliability and long-term performance of the 3D MEMS structure, validating its suitability for practical applications in various industries.

High-technology sockets are in high demand in the semiconductor industry. In this study, a novel 3D MEMS coaxial socket is introduced as a solution. To verify the electrical performance of the proposed socket, a Rhode & Schwarz vector network analyzer (VNA), model ZNA-67, was used to measure the insertion loss of the test vehicle and compared with simulations and measurements up to 60 GHz. Establishing contact with the socket structure for high-speed application poses challenges; thus, different methods such as using a dummy PCB or RF direct probe (GSG type) have been explored. However, dummy PCBs have limitations due to long-length transmission lines and through–via design. Further, RF direct-probe methods face difficulties in achieving stable simultaneous contact on both the top and bottom sides. To overcome these limitations, a 1 mm jack (female) 4 hole.375 square connector from Southwest Microwave, model 2412-01SF, capable of performance up to 110 GHz was employed. The outer dimensions of the connector were 9.52 mm × 9.52 mm, with a length of 9.52 mm and an insertion loss of 0.6 dB at maximum, as specified in the datasheet [23]. This 1 mm connector was adopted to ensure precise alignment and contact in the fabricated dummy vehicle. Additionally, a specialized jig was developed to achieve exact center positioning during measurement, as shown in Figure 13b.

In order to acquire the electrical parameters of the test vehicle, the de-embedding methodology was utilized. De-embedding methods could vary depending on the specific measurement setup and the characteristics of the undesired components. Upon obtaining the transfer function or the response of the undesired components, the mathematical inversion or compensation could be applied to eliminate their effects from subsequent measurements. This ensures a more precise representation of the DUT, as if it were directly measured without the presence of the undesired components. Initially, measurements were conducted of test vehicles equipped with a pair of 1 mm connectors, as well as the pair of 1 mm connectors separately. By obtaining the s-parameter results for each individual component, the de-embedding was then employed to mitigate the influence of the 1 mm connector on the measurements.

Additionally, it was also employed to extract the intrinsic s-parameter values of the DUT, i.e., the socket, by removing the unwanted effects introduced by the 1 mm connector fixture. The measurement setup involved using a VNA to measure the electrical response of the socket. The 1 mm connector, manufactured by Southwest Microwave, was utilized as a fixture in the measurement setup. By applying the de-embedding algorithm [24], the s-parameter values of the socket were obtained, providing insights into its electrical characteristics independent of the connector influence. The insertion loss results obtained using measurement and simulation were compared as shown in Figure 14.

The measured and simulated insertion loss graphs demonstrated a significant correlation, validating the efficacy of the proposed 3D MEMS coaxial socket design. Nevertheless, slight discrepancies were observed particularly above 10 GHz. These variations can be attributed to certain factors that were not completely considered during the simulation phase. One of these factors is contact loss, which was not sufficiently accounted for in the simulation model. Moreover, the contact sensitivity of the 3D MEMS structure exhibited slight deviations from the actual structure owing to variations in the MEMS structure that deviated from the CAD design model.

The measurement setup was highly sensitive, and any vibrations or instability during the measurement process could introduce fluctuations in the s-parameter values. The obtained results were compared with 3D EM simulations conducted using ANSYS HFSS to validate the performance of the proposed socket design.

A comprehensive analysis of the eye diagrams in the time domain was conducted to validate the proposed 3D MEMS coaxial socket. Specifically, the eye diagram of the proposed 3D MEMS coaxial socket model was compared with its corresponding simulation at a data rate of 50 Gb/s for evaluation. A comparison of the eye diagrams at the same data rate is presented in Figure 15, where Figure 15a illustrates the simulated eye diagram of the proposed model and Figure 15b depicts the measured eye diagram. The evaluation focused on parameters such as eye height and eye width (unit interval width), and the results revealed a remarkable similarity between the simulated and measured eye diagrams in the time domain. Finally, the proposed designs of the MEMS structures were successfully verified for the performance of the 3D MEMS coaxial socket in terms of s-parameters up to 60 GHz using the VNA.

## 5. Conclusions

To overcome current challenges in semiconductor package chip testing, a novel 3D MEMS spring structure coaxial socket is proposed as a solution. The socket design specifically addresses the challenges posed by fine-pitch and high-density applications while incorporating impedance matching techniques to ensure optimal performance. To achieve this, a 3D MEMS coaxial test socket with a pitch of 0.6 mm was developed, carefully chosen to match the PUT. To assess the performance of the proposed socket, comprehensive experiments were conducted and analyzed. Further, mechanical tests were conducted to evaluate the structure displacement, ensuring the robustness and reliability of the socket design. Thermal shock cycling tests were also performed to assess the durability and repeatability of the socket under extreme temperatures. Additionally, electrical tests were carried out using a VNA to measure the electrical characteristics of the socket up to 60 GHz, focusing on parameters such as insertion loss and eye diagram.

The results obtained from these experiments reveal the promising performance of the 3D MEMS coaxial socket. The mechanical tests revealed minimal structure displacement, indicating the stability and mechanical integrity of the socket. The thermal shock cycling tests demonstrated the ability of the socket to withstand temperature variations without significant degradation in performance. Furthermore, the electrical tests indicated excellent impedance matching and low insertion loss, confirming the suitability of the socket for high-speed testing applications. The findings of this study highlight the significant potential of the proposed socket design for production and implementation in the semiconductor industry. By addressing the challenges associated with high-technology applications, the 3D MEMS coaxial socket provides a valuable solution to meet the evolving needs of automotive, 5G, and AI devices. Further optimization and refinement of the design will contribute to the advancement of semiconductor testing technology, enabling more efficient and accurate testing processes in the future.

## 6. Patents

The results obtained in this study have been patented in the U.S. (application no.: PK4094229).

## Figures and Tables

**Figure 1 sensors-23-06350-f001:**
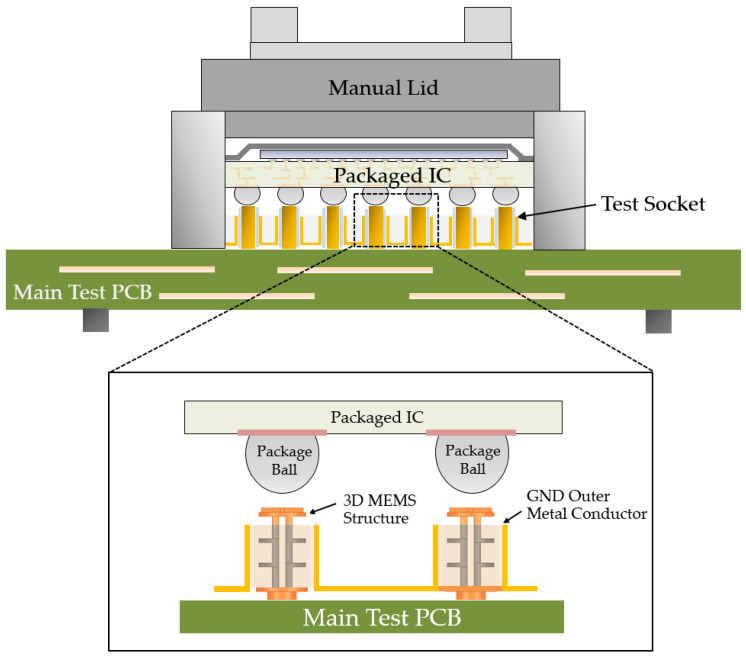
Manual assembly process of a 3D MEMS coaxial socket.

**Figure 2 sensors-23-06350-f002:**
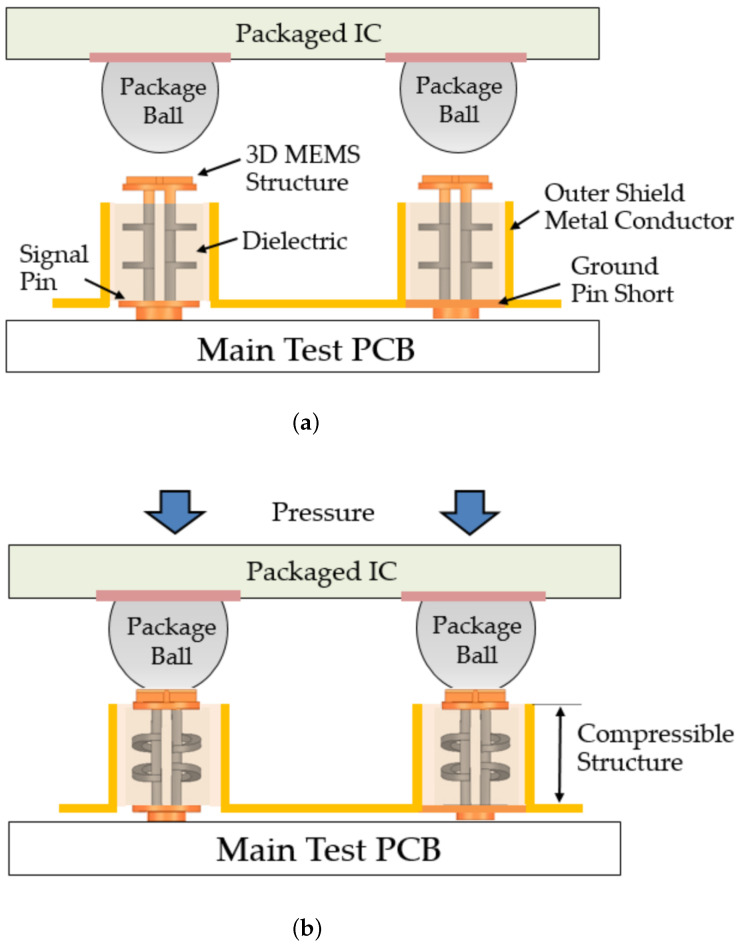
Contact process of a 3D MEMS coaxial structure in the (**a**) normal state and (**b**) testing state.

**Figure 3 sensors-23-06350-f003:**
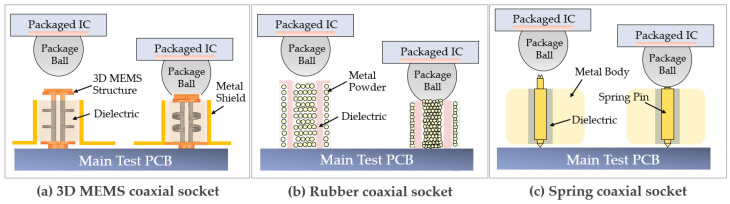
Comparison of three types of coaxial sockets: (**a**) 3D MEMS coaxial socket, (**b**) rubber socket, and (**c**) spring coaxial socket.

**Figure 4 sensors-23-06350-f004:**
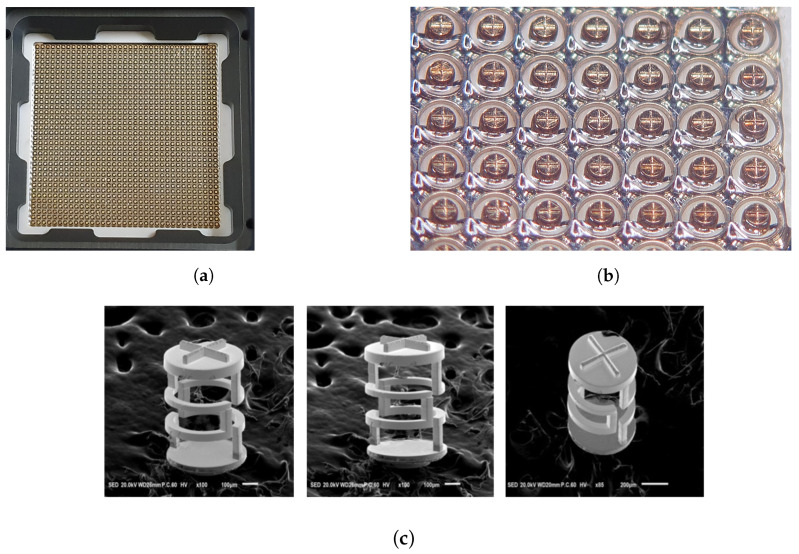
Fabricated 3D MEMS coaxial socket: (**a**) package test socket layout, (**b**) top view of optical microscope, and (**c**) scanning electron microscopy image of the 3D MEMS structure.

**Figure 5 sensors-23-06350-f005:**
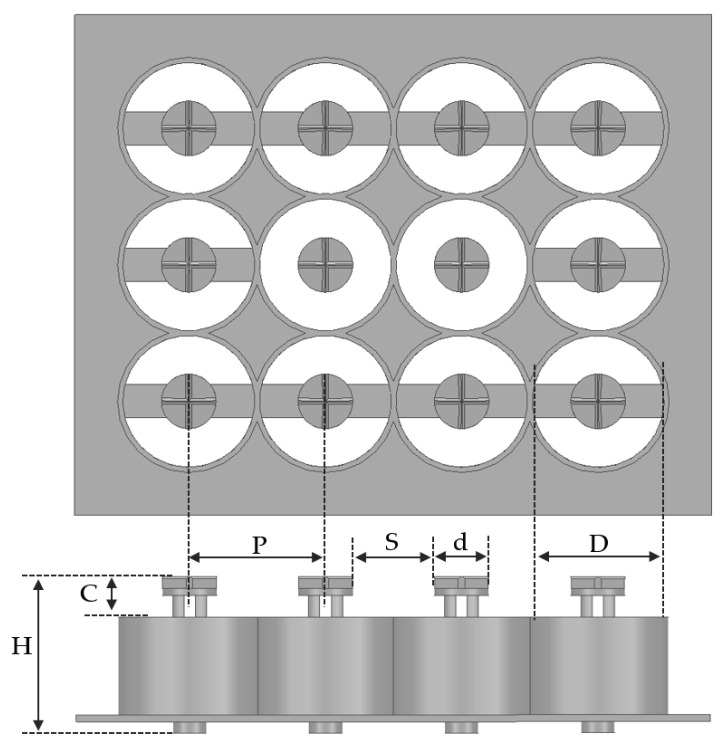
Structure and dimensions of the 3D MEMS coaxial socket.

**Figure 6 sensors-23-06350-f006:**
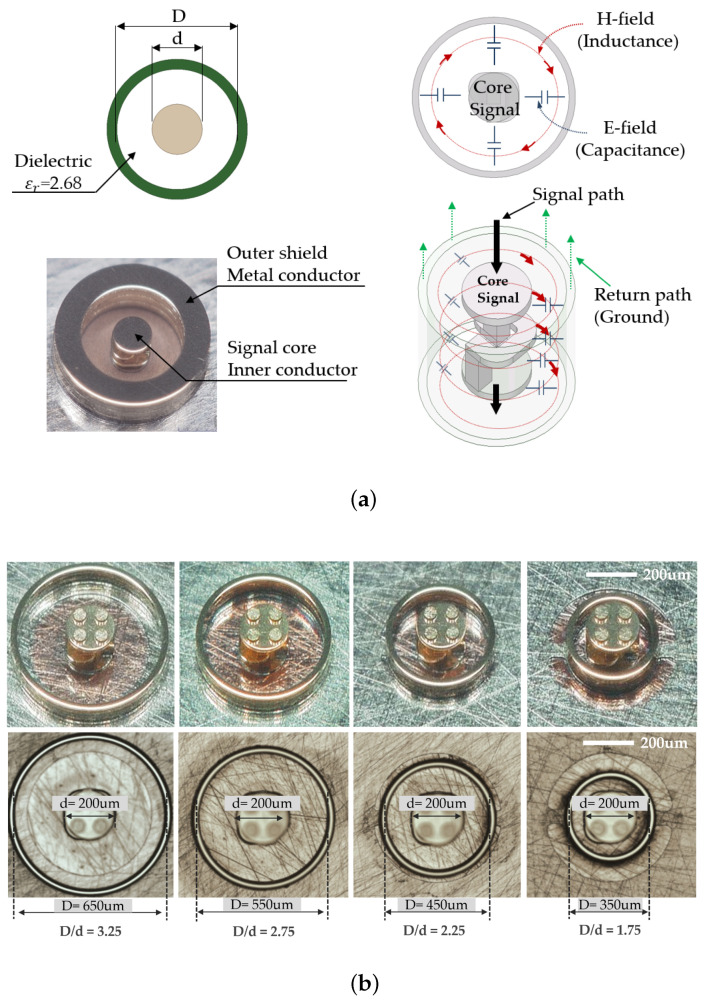
Three-dimensional MEMS structure. (**a**) Illustration of the coaxial geometry and the electric field with the current path. (**b**) Impedance variation due to different outer shield diameter.

**Figure 7 sensors-23-06350-f007:**
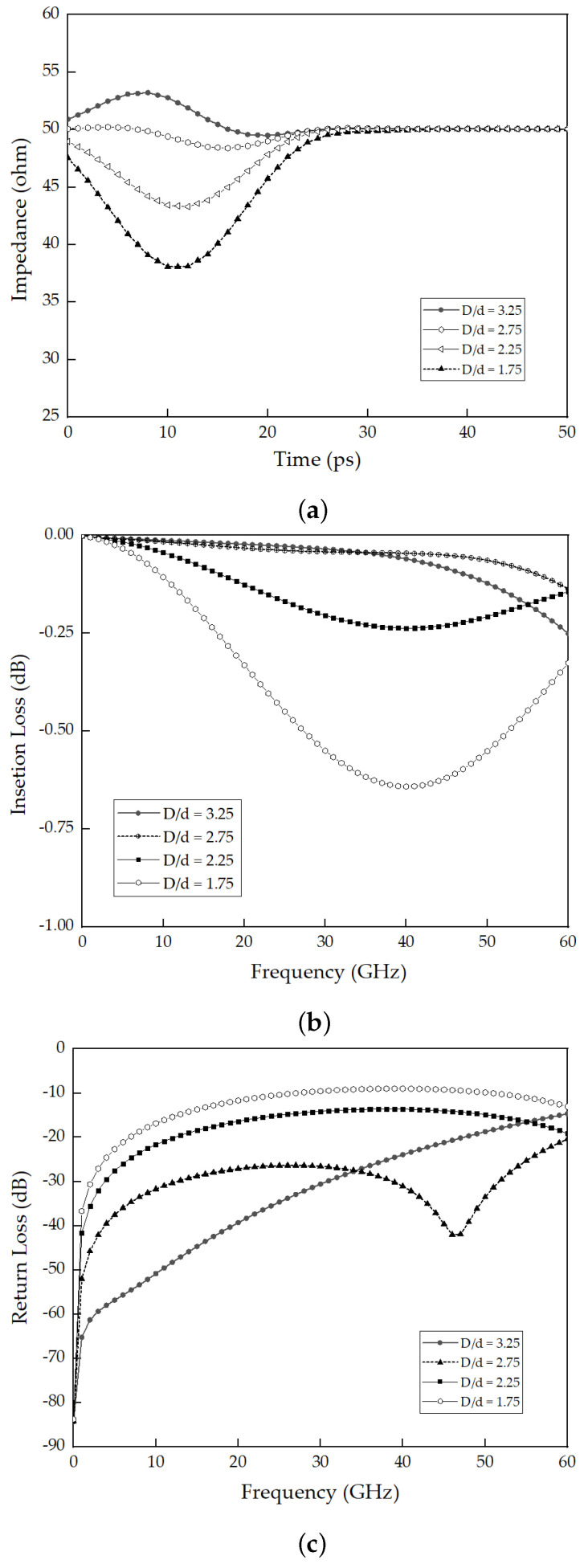
Electrical characteristic parameters of 3D MEMS structure up to 60 GHz varying the inner diameter of the outer conductor, *D/d* (**a**) TDR impedance, (**b**) insertion loss, S21, and (**c**) return loss, S11.

**Figure 8 sensors-23-06350-f008:**
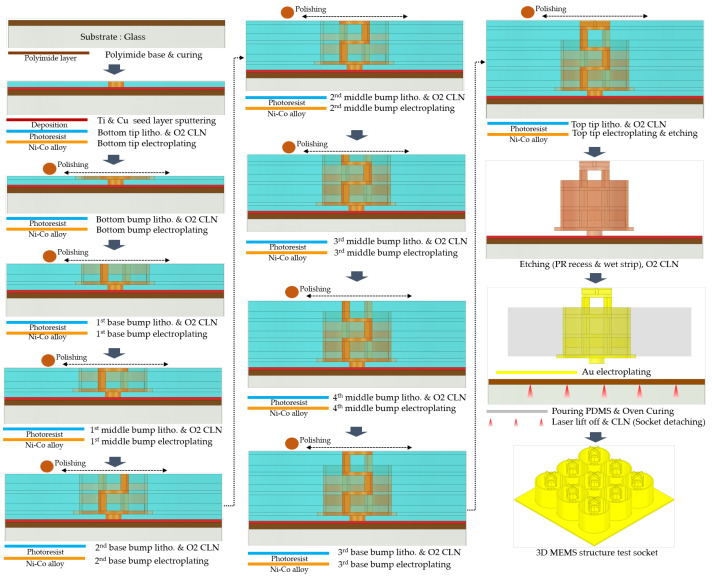
Sequence of the 3D MEMS fabrication process.

**Figure 9 sensors-23-06350-f009:**
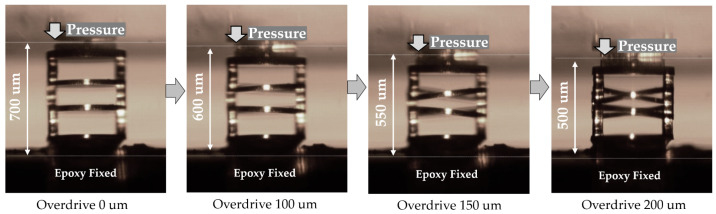
Illustration of 3D MEMS structure under varying overdrive.

**Figure 10 sensors-23-06350-f010:**
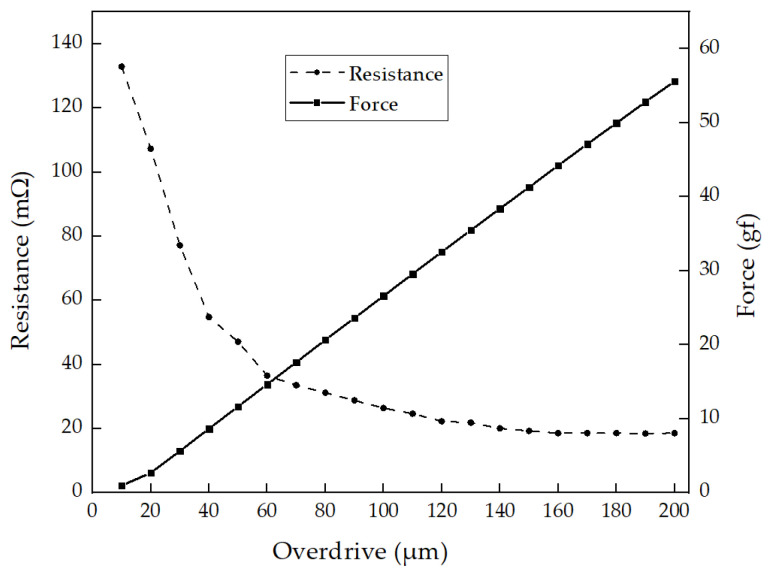
Illustration of structure force and contact resistance under varying overdrive.

**Figure 11 sensors-23-06350-f011:**
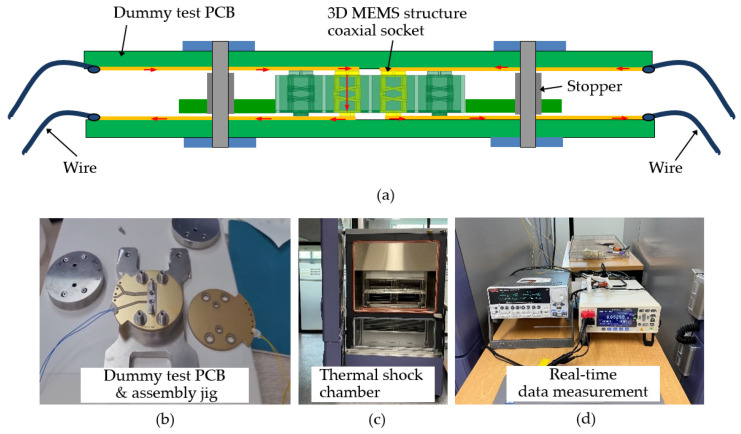
Illustration of the test setup: (**a**) thermal cycle test system, (**b**) dummy test PCB and assembly jig, (**c**) thermal shock chamber, and (**d**) real-time data measurement.

**Figure 12 sensors-23-06350-f012:**
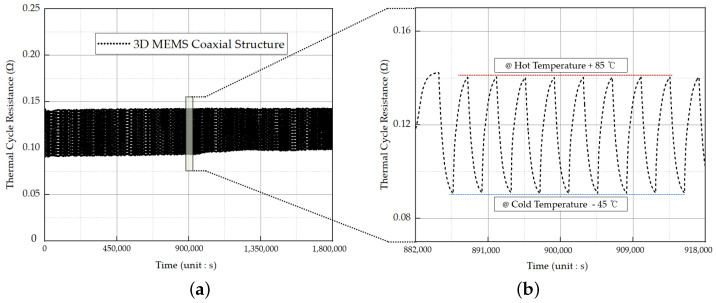
Thermalcycle resistance: (**a**) total value during the period of 21 days and (**b**) value for the period of 36,000 s in low-to-high temperature range.

**Figure 13 sensors-23-06350-f013:**
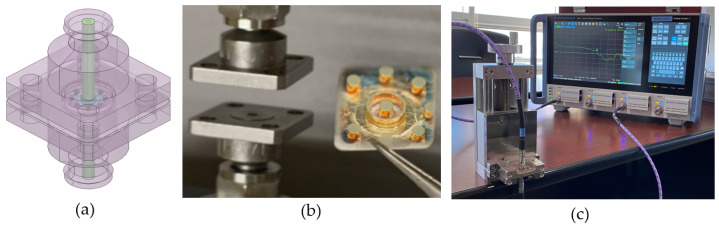
Illustration of test equipment: (**a**) modeling of 1 mm 110 GHz connector, (**b**) 3D MEMS structure test vehicle, and (**c**) network analyzer using measurement jig assembly.

**Figure 14 sensors-23-06350-f014:**
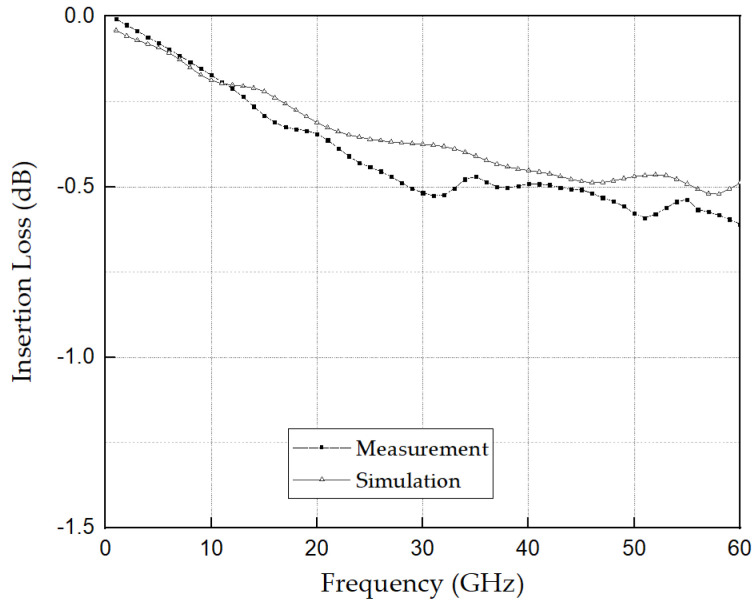
Insertion loss from measurement and 3D EM simulation up to 60 GHz.

**Figure 15 sensors-23-06350-f015:**
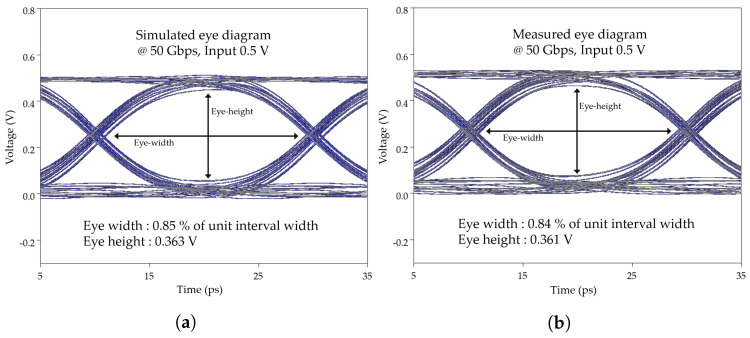
Comparison of the eye diagrams at 50 Gbps: (**a**) simulation and (**b**) measurement.

**Table 1 sensors-23-06350-t001:** Overview for 3 types of coaxial test socket.

	3D MEMS Coaxial Socket	Rubber Coaxial Socket	Spring Coaxial Socket
	· High-speed test		
	· Impedance controllable		
	· Fine pitch (MEMS lithography)	· Manufacturing price	
Pros	· Position accuracy	· Low contact force (less ball damage) [8]	· Reliable contact resistance [7]
	· Large number of pin counts	· Short length (low inductance) [10]	· Long height (long stroke) [5]
	· Stable contact resistance	· Manufacturing time	· sharpened contact tip [7]
	· Durability, longevity		
Cons	· Fabrication difficulty	· Fine-pitch design	· Long length (high inductance)
	· Manufacturing time	· Limited size of coaxial socket	· Machinability [6]
	· Short stroke	· Thermal durability, longevity [8]	· High price
		· Short stroke [9]	

**Table 2 sensors-23-06350-t002:** Physical dimensions of the 3D MEMS coaxial socket.

Symbol	Parameter	Unit	Value
*C*	Head height of the 3D MEMS structure	mm	0.20
*S*	Gap between two heads	mm	0.39
*D*	Inner diameter of the outer conductor	mm	0.58
*d*	Outer diameter of the inner conductor	mm	0.21
*H*	Total height	mm	0.70
*P*	Pad pitch	mm	0.60

**Table 3 sensors-23-06350-t003:** Properties of PDMS (Sylgard 184, Dow Corning).

Property	Unit	Value
Color		Colorless
Mixing	10:1 (Polymer)	
Viscosity (Base)	cP	5100
Viscosity (Mixed)	cP	3500
Thermal Conductivity	W/m ${}^{{}^{\circ}}$K	0.27
Specific Gravity (Cured)		1.03
Cure Time at 25 ${}^{{}^{\circ}}$C	h	48
Heat Cure Time at 100 ${}^{{}^{\circ}}$C	min	35
Heat Cure Time at 125 ${}^{{}^{\circ}}$C	min	25
Heat Cure Time at 150 ${}^{{}^{\circ}}$C	min	10
Durometer Shore		43
Dielectric Strength	kV/mm	19
Volume Resistivity	Ω cm	2.9 × 10^14^
Dielectric Constant at 100 Hz		2.72
Dielectric Constant at 100 kHz		2.68
Dissipation Factor at 100 Hz		0.00257
Dissipation Factor at 100 kHz		0.00133
Linear CTE (by DMA)	ppm/${}^{{}^{\circ}}$C	340
Tensile Strength	MPa	6.7
Useful Temperature Ranges	${}^{{}^{\circ}}$C	−45 to 200
	${}^{{}^{\circ}}$F	−49 to 392

## Data Availability

Not applicable.

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
