# Peer review of "Fabrication and Characterization of Three-Dimensional Microelectromechanical System Coaxial Socket Device for Semiconductor Package Testing"

_sensors, 2023, doi:10.3390/s23146350_

Round 1

Reviewer 1 Report

The paper presents a novel 3D MEMS spring structure coaxial socket for semiconductor package chip testing, which is able to overcome the limitations of formerly proposed technologies utilized for the same purpose (test sockets based on mechanical springs or conductive rubber). The topic is very relevant to the field of packaged IC testing and the solution proposed is inventive, both considering fabrication (3D multilevel metal electroplating embedded within PDMS) and  design. To my knowledge, this is the first MEMS-based solution proposed to the problem addressed, differently than other non-MEMS proposed solutions. The methodology described and the subsequent characterization is fully appropriate in my opinion. The paper is very well written, clear and complete. The solution proposed is inventive and relevant to people working in the MEMS and semiconductor fields. I do not have specific remarks on the paper: I recommend publication in the present form.

Author Response

First of all, thank you for taking your time and consideration of reviewing my manuscript. I have read your comments and I am very grateful for it.

Point 1: The paper presents a novel 3D MEMS spring structure coaxial socket for semiconductor package chip testing, which is able to overcome the limitations of formerly proposed technologies utilized for the same purpose (test sockets based on mechanical springs or conductive rubber). The topic is very relevant to the field of packaged IC testing and the solution proposed is inventive, both considering fabrication (3D multilevel metal electroplating embedded within PDMS) and  design. To my knowledge, this is the first MEMS-based solution proposed to the problem addressed, differently than other non-MEMS proposed solutions. The methodology described and the subsequent characterization is fully appropriate in my opinion. The paper is very well written, clear and complete. The solution proposed is inventive and relevant to people working in the MEMS and semiconductor fields. I do not have specific remarks on the paper: I recommend publication in the present form.

I appreciate your comments and hope to publish my manuscript as well.

Reviewer 2 Report

The manuscript demonstrated a microfabricated coaxial socket device for semiconductor testing, which showed some advantages over existing technologies. The results are quite promising.  However, before it can be published, more detailed design and testing results are expected. Here are my suggestions:

1. The design process is lack in details. From line 94, the authors claimed they did thorough evaluations by using FEM simulation. How did it lead to the final design? More detailed procedures and analyses will be beneficial to the reads.

2.  The description of fabrication process is not informative enough. The process flow chart is also not very clear, i.e., no corresponding legend. The readers could have difficulties to follow the microfabrication process. Please provide more details.

3. The thermal cycle testing results demonstrated the stability of the proposed socket structures. I am wondering how dose it perform under force load cycles? More testing results and/or explanations are expected.

Reviewer 3 Report

Before considering the publications, there are several important points are missing in the manuscript.

On page 3, authors have mentioned that “This advanced socket solution provides distinct advantages over conventional sockets, including precise impedance control, high-frequency signal quality, shortened signal-transmission paths, customizable design capabilities for high-density and fine-pitch applications, enhanced durability and longevity, and compatibility with a wide range of packaged ICs.” Authors should explain why and how, for instance should give the precise impedance control, 0.1 Ohm ? or 1 Ohm at which frequency ? besides, high-frequency signal quality… what is the frequency range ? …etc.

For making comparisons of the three kinds of sockets shown in Fig.3, it is better to make a table and list all parameters which will be clearer for readers.

As discussed by the authors, impedance match is crucial. Therefore, please provide the HFSS simulation results for impedance, e.g. S11 and S21 simulated in a large frequency range. These two parameters are essential for discussions and understanding of impedance.

Figure 7 is very confusing, including the explanations on page-7. E.g. “The process began by creating the base layer and performing lithography to define the desired pattern. Subsequently, an etching process was employed to clean the surface, followed by electroplating. This sequence was repeated for the first and second middle bumps. Electrodeposition offers the advantage of producing deposits with customized structures.”. What is the base layer? Which substrate? How to etch? dry or we etching ? What is the selectivity of the etching process. ? “Electrodeposition” ? through which methods? Evaporation or sputtering? What is the metal?... These details are very important for readers to see the difficulties or feasibilities of fabrications. 

Reviewer 4 Report

The authors proposed a configuration of a MEMS socket for semiconductor testing. The study is interesting and I believe it suited the interest of the readers of this journal. The manuscript is generally well written, just several concerns prior to publication.

Please include a scale bar in Fig. 6 and 8 for better illustration of the device. It is also not so easy to follow the fabrication sequence described in Fig. 7. It would be helpful if there is a color-material index, and a guide in the explanation about which process is being explained.

The caption of Fig. 11 seems not to be as intended. And for the rest is fine.

Generally fine and understandable.

Round 2

Reviewer 2 Report

The revised manuscript has addressed my concerns. I recommend it to be published at Sensors.